# Dual Learning: Theoretical Study and Algorithmic Extensions

## Abstract

Dual learning has been successfully applied in many machine learning applications including machine translation, image-to-image transformation, etc. The high-level idea of dual learning is very intuitive: if we map an $x$ from one domain to another and then map it back, we should recover the original $x$. Although its effectiveness has been empirically verified, theoretical understanding of dual learning is still missing. In this paper, we conduct a theoretical study to understand why and when dual learning can improve a mapping function. Based on the theoretical discoveries, we extend dual learning by introducing more related mappings and propose highly symmetric frameworks, cycle dual learning and multipath dual learning, in both of which we can leverage the feedback signals from additional domains to improve the qualities of the mappings. We prove that both cycle dual learning and multipath dual learning can boost the performance of standard dual learning under mild conditions. Experiments on WMT 14 English↔German and MultiUN English↔French translations verify our theoretical findings on dual learning, and the results on the translations among English, French, and Spanish of MultiUN demonstrate the efficacy of cycle dual learning and multipath dual learning.

## 1 Introduction

Most machine learning tasks can be formulated as learning a mapping from one domain to another one, like image classification (from image to label), neural machine translation (from source language to target language), speech recognition (from voice to text), etc. Among them, quite a few tasks are of dual forms, like image classification v.s. image generation (from label to image), the neural machine translation between two languages (English→French v.s. French→English), speech recognition v.s. speech synthesis (from text to voice), etc. Such duality can be utilized to improve the model qualities.

One prominent framework is dual learning, first proposed by He et al. (2016) for machine translation and then applied to many other applications like image translation (Kim et al., 2017; Zhu et al., 2017), question answering and generation (Tang et al., 2017), etc. In dual learning, two mapping functions between two domains are trained simultaneously so that one function is close to the inverse of the other. The intuition is that, if we translate a sentence from English to French and then translate the obtained French sentence back to English, we should get the same sentence or a very simlar one.

Dual learning is of great interest because it can accommodate any unidirectional architecture, e.g. a transformer (Vaswani et al., 2017), and provide a performance boost. Moreover, dual learning can be used in semi-supervised learning, which is highly desirable since deep neural networks are generally thirst for labeled data.

Despite of the empirical success of dual learning, the theoretical understanding is largely missing. In this paper, we conduct both theoretical analyses and empirical studies to answer the following questions:
**Q1:** Why and when does dual learning improve a mapping function?
**Q2:** Can we further improve the performance of a mapping function?

### 1.1 OUR CONTRIBUTIONS

Our contributions are in two folds: a theoretical study of dual learning and the frameworks of cycle dual learning and multipath dual learning, which subsume dual learning as a special case. Without loss of generality, we take machine translation as the example for the study and algorithm presentation.

**Dual learning.** We take a novel statistical approach to model the problem. Suppose there are two baseline translators between two language spaces, one forward and one backward. Based on our Theorem 1, dual learning outperforms both baseline translators as long as the baseline translators are better than random mappings under natural assumptions. Empirical studies show that an improvement is observed even if the reconstruction is far from perfect.

**Cycle dual learning and multipath dual learning.** We propose cycle dual learning and multipath dual learning frameworks by extending dual learning. Both frameworks use dual learning as the basic building block and leverage a third, a fourth, or more languages to help boost the translator qualities between the original two languages. We prove that under mild conditions, the two symmetric frameworks outperform dual learning (Theorems 2 and 3). Our experiments on MultiUN dataset show a significant improvement (1.45 BLEU points, see Table 4) against dual learning.

### 1.2 RELATED WORK

Dual learning was first proposed by He et al. (2016) in the context of machine translation, where the two dual translators are updated in a reinforcement learning manner with the reconstructed distortion as the feedback signal. A similar approach proposed by Cheng et al. (2016) has the same high-level idea but a very different implementation. Since then dual learning architectures have been proposed for other applications including image processing (Kim et al., 2017; Zhu et al., 2017), sentiment analysis (Xia et al., 2017a), image segmentation (Luo et al., 2017), etc.

Built upon the dual learning framework, Xia et al. (2017b) and Wang et al. (2018) consider the joint distribution constraint, which says the joint distribution of sample over two domains is invariant when computing from either domain. We relax this constraint for simplicity of analysis. Xia et al. (2018) proposed model-level dual learning, which shares components between the primary direction and the dual direction. Dual learning was also leveraged for unsupervised learning (Lample et al., 2018; Artetxe et al., 2018).

Despite the vast number of works related to dual learning, none of them explain why it works or when it fails. Galanti et al. (2018) claim that dual learning does not circumvent the alignment problem, where a sentence is translated wrong by the forward translator but translated back to it by the backward translator. We show that the alignment problem occurs with a small probability under dual learning compared to the baseline translator, and this probability can be further reduced by our cycle dual learning and multipath dual learning. Furthermore, their hypothesis that the translator should not be too complex is not verified in the context of machine translation.

## 2 PRELIMINARIES

In this section, we introduce the notations and the essential assumptions used for theoretical analysis. A table summarizing the notations can be found in Appendix A. Let $S_1, \ldots, S_k$ be $k$ language spaces, composed of sentences in each language. For any $S_i$, we denote the distribution of sentences in $S_i$ by $\mu^{(i)}$ and let $X^{(i)}$ be the random variable, i.e. $\Pr(X^{(i)} = x) = \mu^{(i)}(x)$. For any $x^{(i)} \in S_i$, we denote the correct translation in $S_j$ by $x^{(j)}$. In practice, we may select a threshold BLEU (Papineni et al., 2002a) score, above which the translation is considered correct. Let $T_{ij}$ denote the baseline translator that translates from a sentence in $S_i$ to one in $S_j$, i.e. the desired mapping is $T_{ij}(x^{(i)}) = x^{(j)}$. When a random sentence $x^{(i)}$ is sampled from $S_i$ according to $\mu^{(i)}$, it is possible that this sentence is translated incorrectly. We use accuracy of the translator $p_{ij}$, the accuracy of the translator, to describe the probability of translating a sentence correctly when this sentence is randomly sampled from $S_i$ according to $\mu^{(i)}$. Formally, $p_{ij} = \Pr_{x^{(i)} \sim \mu^{(i)}}(T_{ij}(x^{(i)}) = x^{(j)}) = \mathbb{E}_{x^{(i)} \sim \mu^{(i)}}[\mathbb{1}_{T_{ij}(x^{(i)})=x^{(j)}}] = \sum_{x^{(i)} \in S_i} \mathbb{1}_{T_{ij}(x^{(i)})=x^{(j)}} \mu^{(i)}(x^{(i)})$, where $\mathbb{1}$ is the indicator function.

We sometimes omit the subscript $x^{(i)} \sim \mu^{(i)}$ for simplicity. It is also easy to see that

$$\Pr(T_{ij}(x^{(i)}) \neq x^{(j)}) = \mathbb{E}[\mathbb{1}_{T_{ij}(x^{(i)}) \neq x^{(j)}}] = 1 - p_{ij}. \tag{1}$$

In order to characterize reconstruction accuracy, we let $X^{(j,r)}$ denote the random variable that follows the distribution $T_{ij}(X^{(i)})$ where $X^{(i)} \sim \mu^{(i)}$. We define $\mu^{(j,r)}(x) = \Pr(X^{(j,r)} = x)$, and further define $p_{ji}^r = \Pr_{X^{(j,r)} \sim \mu^{(j,r)}}(T_{ji}(x^{(j)}) = x^{(i)}) = E_{X^{(j,r)} \sim \mu^{(j,r)}}[\mathbb{1}_{T_{ji}(x^{(j)}) = x^{(i)}}] = \sum_{x^{(j)} \in S_j} \mathbb{1}_{T_{ji}(x^{(j)}) = x^{(i)}} \mu^{(j,r)}(x^{(j)})$. The superscript $r$ means "reconstruction". The difference between $p_{ji}^r$ and $p_{ji}$ lies in the distributions of samples in space $S_j$. We assume that $T_{ij}$ and $T_{ji}$ are independent translators. This is natural because $T_{ij}$ and $T_{ji}$ are usually trained independently. Then, for any $j \neq i$, we have

$$\Pr(T_{ij}(x^{(i)}) = x^{(j)}, T_{ji}(x^{(j)}) = x^{(i)}) = p_{ij} p_{ji}^r. \tag{2}$$

Further, we make the following assumption.

**Assumption 1.** *For any $j \neq i$, any $y^{(j)} \in S_j$, $E[\mathbb{1}_{y^{(j)} \neq x^{(j)}, T_{ij}(x^{(i)}) = y^{(j)}}] = \frac{1 - p_{ij}}{m}$, where $m$ is a constant.*

The interpretation of this assumption is as follows. For all $x^{(i)} \in S_i$ that are incorrectly translated into $S_j$, there are $m$ possibly incorrect translations (in $S_j$) with positive probabilities. Assumption 1 states that each of the incorrect translations is chosen uniformly at random.

**Justification of Assumption 1.** It is generally not true in the real-world since the probabilities of different translations are usually very different. However, this assumption can be approximated by partitioning the possibly wrong translations into several clusters, where the probability of the translation falling into each cluster is close. The best case is that the incorrect translation with the largest probability $p_{\max}$ is treated as one cluster. All the other possibly incorrect translations are clustered so that the probability of each cluster is close to $p_{\max}$. Then we have $m \leq \frac{1 - p_{ij}}{p_{\max}}$. A small $p_{\max}$ (a large $m$) is desirable. Note that the correct translation is indeed also a group of translations if we set a threshold BLEU score, above which the translation is considered correct.

To analyze the efficacy of dual learning, we consider the following two cases:
$$\text{(Case 1) } T_{ji}(T_{ij}(x^{(i)})) = x^{(i)}; \qquad \text{(Case 2) } T_{ji}(T_{ij}(x^{(i)})) \neq x^{(i)}.$$

Dual learning will detect Case 2 and train the translators so that Case 2 is minimized. We use superscript $d$ for the translators trained from dual learning. For any given $x_i \in S_i$ which fails in reconstruction (Case 2), we define

$$\alpha = \Pr(T_{ij}^d(x^{(i)}) = x^{(j)}, T_{ji}^d(T_{ij}^d(x^{(i)})) = x^{(i)} | \text{Case 2}) \tag{3}$$

$$\beta = \Pr(T_{ij}^d(x^{(i)}) \neq x^{(j)}, T_{ji}^d(T_{ij}^d(x^{(i)})) = x^{(i)} | \text{Case 2}) \tag{4}$$

$$\gamma = \Pr(T_{ji}^d(T_{ij}^d(x^{(i)})) \neq x^{(i)} | \text{Case 2}), \tag{5}$$

where "Case 2" denotes the condition $T_{ji}(T_{ij}(x^{(i)})) \neq x^{(i)}$. Here $\alpha$ can be viewed as the probability of correcting the wrong translations by dual learning, $\beta$ the probability of the occurrence of the alignment problem under Case 2, and $\gamma$ the probability of nonzero reconstruction error. $\gamma$ models the imperfectness of dual learning, which should be zero in the ideal case. It is easy to see $\alpha + \beta + \gamma = 1$.

## 3 THEORETICAL STUDY

In this section, we provide a theoretical study of why dual learning outperforms the baseline translator by the following theorem.

**Theorem 1.** *Under Assumption 1, for any language spaces $S_i$ and $S_j$, the accuracy of dual learning outcome $T_{ij}^d$ is $p_{ij}^d = (1 - \alpha) p_{ij} p_{ji}^r + \alpha(1 - \frac{(1 - p_{ij})(1 - p_{ji}^r)}{m})$, where $\alpha$ is defined in equation 3.*

This theorem is proved by explicitly computing the probabilities of each case. During dual learning, Case 2 is redistributed to $\alpha$ case (a desirable subcase of Case 1, defined in equation 3), $\beta$ case (an undesirable but unavoidable subcase of Case 1, defined in equation 4), and $\gamma$ case (remaining in Case 2, defined in equation 5). See Appendix B for the full proof. We have the following observations from Theorem 1.

**Relation to the baseline translators.** The accuracy is dual learning improvement is positively correlated to the baseline translators of both directions. The larger the $p_{ij}$ or $p_{ji}^r$ is, the higher accuracy of $T_{ij}^d$ dual learning can achieve.

**The role of $m$ and $\alpha$.** When $p_{ij}, p_{ji}^r$ are fixed, a large $m$ is desired for better accuracy. As $\alpha$ increases, the outcome of dual learning improves. In the optimistic (but maybe unrealistic) case where $\alpha = 1, \beta = \gamma = 0$, we have $p_{ij}^d - p_{ij} = 1 - \frac{(1-p_{ij})(1-p_{ji}^r)}{m} - p_{ij} = \frac{(1-p_{ij})(m-1+p_{ji}^r)}{m}$. This means dual learning provides an improvement as long as $m \geq 1$ and $p_{ji}^r > 0$, which is trivial.

**A hypothesis.** We consider the case where the probabilities of redistribution to $\alpha$ case and $\beta$ case are proportional to $\alpha$ and $\beta$. Formally, $\frac{\alpha}{\beta} = \frac{\Pr(T_{ij}(x^{(i)})=x^{(j)}, T_{ji}(T_{ij}(x^{(i)}))=x^{(i)})}{\Pr(T_{ij}(x^{(i)}) \neq x^{(j)}, T_{ji}(T_{ij}(x^{(i)}))=x^{(i)})} = \frac{mp_{ij}p_{ji}^r}{(1-p_{ij})(1-p_{ji}^r)}$. Then we have $\alpha = \frac{mp_{ij}p_{ji}^r(1-\gamma)}{mp_{ij}p_{ji}^r+(1-p_{ij})(1-p_{ji}^r)}$ and

$$
\begin{aligned}
p_{ij}^d &= \Pr(T_{ij}^d(x^{(i)}) = x^{(j)}, T_{ji}^d(x^{(j)}) = x^{(i)}) \\
&= \frac{mp_{ij}p_{ji}^r(1 - \gamma(1 - p_{ij}p_{ji}^r - \frac{(1-p_{ij})(1-p_{ji}^r)}{m}))}{mp_{ij}p_{ji}^r + (1 - p_{ij})(1 - p_{ji}^r)} = \frac{mp_{ij}p_{ji}^r(1 - \Gamma)}{mp_{ij}p_{ji}^r + (1 - p_{ij})(1 - p_{ji}^r)},
\end{aligned} \quad (6)
$$

where $\Gamma = \gamma(1 - p_{ij}p_{ji}^r - \frac{(1-p_{ij})(1-p_{ji}^r)}{m})$. To compare $p_{ij}^d$ with the accuracy of the original translator, we compute the difference

$$
\begin{aligned}
p_{ij}^d - p_{ij} &= p_{ij}\left(\frac{mp_{ji}^r(1 - \Gamma)}{mp_{ij}p_{ji}^r + (1 - p_{ij})(1 - p_{ji}^r)} - 1\right) \\
&= p_{ij}\left(\frac{mp_{ji}^r}{mp_{ij}p_{ji}^r + (1 - p_{ij})(1 - p_{ji}^r)} - 1 - \frac{\Gamma mp_{ji}^r}{mp_{ij}p_{ji}^r + (1 - p_{ij})(1 - p_{ji}^r)}\right) \\
&= p_{ij}\left(\frac{(1 - p_{ij})((m + 1)p_{ji}^r - 1)}{mp_{ij}p_{ji}^r + (1 - p_{ij})(1 - p_{ji}^r)} - \frac{\Gamma mp_{ji}^r}{mp_{ij}p_{ji}^r + (1 - p_{ij})(1 - p_{ji}^r)}\right)
\end{aligned}
$$

Ideally, we have $\gamma = 0$, which means $\Gamma = 0$. If $p_{ji}^r > \frac{1}{m+1}$, the outcome of dual learning is better than the baseline translator. This means, as long as the overall probability of a correct translation is greater than choosing from the $m + 1$ translations (1 correct and $m$ incorrect) uniformly at random, dual learning outperforms the baseline translator. The expression with the $\Gamma$ factor is negative, which is consistent with the intuition that $\gamma$ should be minimized. For the nonideal case where $\Gamma > 0$, if both $p_{ji}^r > \frac{1}{m+1}$ and $\Gamma < (1 - p_{ij})(1 + \frac{1}{m} - \frac{1}{mp_{ji}^r})$ hold, dual learning also improves against the baseline translator.

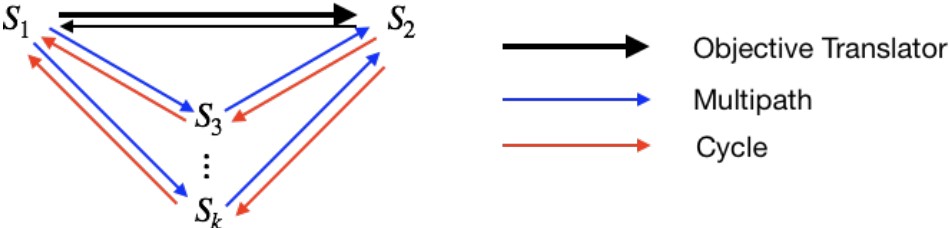

Figure 1: The proposed multipath dual learning and cycle dual learning.

# 4 EXTENSIONS: CYCLE DUAL LEARNING AND MULTIPATH DUAL LEARNING

In Theorem 1, we found that both $p_{ij}$ and $p_{ji}^r$ play a positive role in improving $p_{ij}^d$ under mild assumptions. A natural question is whether this probability could be further enhanced by exploiting multiple language domains. Therefore, we propose the two frameworks, cycle dual learning and multipath dual learning, both of which leverage multiple languages and significantly extend the standard dual learning. In this section, we mainly present the cycle dual learning framework due to the space constraint.

## 4.1 FRAMEWORK

The proposed frameworks are illustrated in Figure 1. Let $S_1$ and $S_2$ denote the source language space and the target language space respectively. To use these frameworks, we first train the following translators: $S_1 \leftrightarrow S_2$, $S_1 \leftrightarrow S_k$ and $S_2 \leftrightarrow S_k$ where $k \geq 3$. Then, for cycle dual learning, we require a sentence from $S_2$ to be very similar to $S_2 \rightarrow S_k \rightarrow S_1 \rightarrow S_2$ (or equivalently, a sentence from $S_1$ to be very similar to $S_1 \rightarrow S_2 \rightarrow S_k \rightarrow S_1$); for multipath dual learning, we require the translation $S_1 \rightarrow S_2$ to be the very similar $S_1 \rightarrow S_k \rightarrow S_2$. In this way, we build another constraint, where the translation $S_1 \rightarrow S_2$ could leverage the information pivoted by the domain $S_k$. In practice, to use cycle dual learning to enhance the $S_1 \rightarrow S_2$ model, we need to minimize $\sum_{x^{(2)} \in S_2} D(T_{12}(T_{k1}(T_{2k}(x^{(2)}))), x^{(2)})$, where $D(\cdot, \cdot)$ measures the differences of two inputs; to use multipath dual learning, we need to minimize $\sum_{x^{(1)} \in S_1} D(T_{12}(x^{(1)}), T_{k2}(T_{1k}(x^{(1)})))$. Similar update could also be applied to $S_2 \rightarrow S_1$ translation and leverage more language domains. If no auxiliary domain is provided, both cycle dual learning and multipath dual learning will degenerate to the standard dual learning. We design sampling based algorithms for the two frameworks. This section focuses on the cycle dual learning framework with corresponding theoretical analysis. See Appendix D and F for the multipath dual learning framework. In both frameworks, let $\theta_{ij}$ denote the parameters of translator $T_{ij}$.

---

**Algorithm 1:** Cycle Dual Learning Framework

---

1 *Input*: Samples from spaces $S_1 \ldots S_k$, initial translators $T_{12}, T_{21}$ and $T_{1i}, T_{i1} \; \forall i = 3, \ldots, K$; learning rates $\eta$;
2 Train each of $T_{12}, T_{21}$ and $T_{1i}, T_{i1} \; \forall i = 3, \ldots, k$ by dual learning;
3 Randomly sample a $k$ from $\{3, 4, \cdots, K\}$; randomly sample one $x^{(1)} \in S_1$ and one $x^{(2)} \in S_2$;
4 Generate $\tilde{x}^{(2)}$ by $T_{k2} \circ T_{1k}(x^{(1)})$ and generate $\tilde{x}^{(1)}$ by $T_{k1} \circ T_{2k}(x^{(2)})$;
5 Update the parameters of $T_{12}$ and $T_{21}$, denoted as $\theta_{12}$ and $\theta_{21}$, as follows:

$$\theta_{12} \leftarrow \theta_{12} + \eta \nabla_{\theta_{12}} \log P(x^{(2)}|\tilde{x}^{(1)}; \theta_{12}); \quad \theta_{21} \leftarrow \theta_{21} + \eta \nabla_{\theta_{21}} \log P(x^{(1)}|\tilde{x}^{(2)}; \theta_{21}); \quad (7)$$

Repeat Step 3 to Step 5 until convergence;

---

## 4.2 THEORETICAL ANALYSIS

The theoretical analyses of cycle dual learning and multipath dual learning are highly symmetric. In this section, we take cycle dual learning as an example. The analysis for multipath dual learning can be found in Appendix F.

For simplicity, we focus on the triangle structure that contains only $S_1$, $S_2$ and $S_3$. Now dual learning serves as the basic structure. For simplicity, we use $q_{ij}$ to denote the accuracy of $T_{ij}^d$, i.e. $q_{ij} = p_{ij}^d$. We further define $q_{31}^{(2)} = \Pr_{X^{(3)} \sim T_{23}^d(X^{(1)})}(T_{31}^d(x^{(3)}) = x^{(1)})$, which as the accuracy of $T_{31}^d$ when the sentence in $S_3$ is translated from a random sentence in $S_2$. This is analogous to reconstruction accuracy. To analyze cycle dual learning, we consider the following two cases (analogous to Case 1 and Case 2 for dual learning):
**Case 1:** $x^{(1)} = T_{31}^d(T_{23}^d(T_{12}^d(x^{(1)})))$; and **Case 2:** $x^{(1)} \neq T_{31}^d(T_{23}^d(T_{12}^d(x^{(1)})))$.

Cycle dual learning will detect Case 2 and train the translators so that Case 2 is minimized. To quantify this effect, we define the following probabilities:

$$\alpha' = \Pr(T_{12}^c(x^{(1)}) = x^{(2)}, T_{31}^c(T_{23}^c(x^{(2)})) = x^{(1)}|\text{Case 2})$$
$$\beta' = \Pr(T_{12}^c(x^{(1)}) \neq x^{(2)}, T_{31}^c(T_{23}^c(T_{12}^c(x^{(1)}))) = x^{(1)}|\text{Case 2})$$
$$\gamma' = \Pr(x^{(1)} \neq T_{31}^c(T_{23}^c(T_{12}^c(x^{(1)})))|\text{Case 2}), \quad (8)$$

where Case 2 denotes the condition $x^{(1)} \neq T_{31}^d(T_{23}^d(T_{12}^d(x^{(1)})))$ and superscripts $d$ and $c$ denote the translators obtained from dual learning and cycle dual learning, respectively. $\alpha', \beta', \gamma'$ can be viewed as the probability of correcting the wrong translations by cycle dual learning, the probability of the occurrence of the alignment problem under Case 2, and the probability of nonzero reconstruction

error. $\gamma'$ models the imperfectness of dual learning. And we have $\alpha' + \beta' + \gamma' = 1$. We assume that $T_{23}^d$ and $T_{31}^d$ are independent translators. That implies,

$$\Pr(T_{23}^d(x^{(2)}) = x^{(3)}, T_{31}^d(x^{(3)}) = x^{(1)}) = q_{23}q_{31}^{(2)} \tag{9}$$

We will also need a similar assumption to Assumption 1 as follows.

**Assumption 2.** *For any $j \neq i$, any $y^{(j)} \in S_j$, $E[\mathbb{1}_{y^{(j)} \neq x^{(j)}, T_{ij}^d(x^{(i)}) = y^{(j)}}] = \frac{1-q_{ij}}{m}$, where $m$ is a constant.*

The interpretation and justification of Assumption 2 is similar to those of Assumption 1. Then we have the following theorem about the triangle structure. The more general case can be viewed as adding one path a time so Theorem 2 can be applied.

**Theorem 2.** *Given languages spaces $S_1$, $S_2$, and $S_3$, where the objective is to train a translator that maps from $S_1$ to $S_2$, the accuracy of cycle dual learning outcome is*

$$p_{12}^c = q_{12}(1 - \alpha')(q_{23}q_{31}^{(1)} + \frac{(1-q_{23})(1-q_{31}^{(1)})}{m})$$
$$+ \alpha'(1 - \frac{(1-q_{12})(q_{23} + q_{31}^{(1)} - (m+1)q_{23}q_{31}^{(1)} + m - 1)}{m^2}) \tag{10}$$

*under Assumption 2.*

Theorem 2 is also proved by explicitly computing the probability of each case. The full proof can be found in the Appendix C. Theorem 2 is much harder to interpret than Theorem 1, but we can still observe how factors affect the accuracy of $T_{12}^c$.

**The role of $T_{12}^d$.** The accuracy increases as $q_{12}$ increases. So a good initial translator is desirable. However, if $T_{12}^d$ is arbitrarily bad, it is still possible to improve it significantly. This implies the application of this framework on low-resource or even zero-shot machine translation.

**A similar hypothesis.** Similar to the analysis of dual learning, we consider the condition where $\frac{\alpha'}{\beta'} = \frac{\Pr(T_{12}^d(x^{(1)}) = x^{(2)}, T_{31}^d(T_{23}^d(x^{(2)})) = x^{(1)})}{\Pr(T_{12}^d(x^{(1)}) \neq x^{(2)}, T_{31}^d(T_{23}^d(T_{12}^d(x^{(1)}))) = x^{(1)})}$ and define $\Gamma' = \gamma' \Pr(x^{(1)} \neq T_{31}^d(T_{23}^d(T_{12}^d(x^{(1)}))))$. Then the accuracy simplifies to

$$p_{12}^c = \frac{\alpha'(1-\Gamma')}{\alpha' + \beta'} = \frac{q_{12}(q_{23}q_{31}^{(2)} + \frac{(1-q_{23})(1-q_{31}^{(2)})}{m})(1-\Gamma')}{q_{12}(q_{23}q_{31}^{(2)} + \frac{(1-q_{23})(1-q_{31}^{(2)})}{m}) + \frac{(1-q_{12})(q_{23}+q_{31}^{(2)}-(m+1)q_{23}q_{31}^{(2)}+m-1)}{m^2}}$$
$$= \frac{1-\Gamma'}{1 + \frac{(1-q_{12})(q_{23}+q_{31}^{(2)}-(m+1)q_{23}q_{31}^{(2)}+m-1)}{mq_{12}(mq_{23}q_{31}^{(2)}+(1-q_{23})(1-q_{31}^{(2)}))}} = \frac{1-\Gamma'}{1 + M\frac{1-q_{12}}{q_{12}}}$$

where $M = \frac{q_{23}+q_{31}^{(2)}-(m+1)q_{23}q_{31}^{(2)}+m-1}{m(mq_{23}q_{31}^{(2)}+(1-q_{23})(1-q_{31}^{(2)}))}$. We observe that When $\gamma' = 0$ (and therefore $\Gamma' = 0$) and $M = 1$, it simplifies to $q_{12}$, which is the accuracy of dual learning and that $p_{12}^c$ increases as $M$ decreases. To characterize the condition when $p_{12}^c > q_{12}$, we let $M < 1$. We have $\frac{q_{13}+q_{32}^{(1)}-(m+1)q_{13}q_{32}^{(1)}+m-1}{m(mq_{13}q_{32}^{(1)}+(1-q_{13})(1-q_{32}^{(1)}))} < 1$, which leads to

$$((m+1)q_{13} - 1)((m+1)q_{32}^{(1)} - 1) > 0.$$

When $q_{13}, q_{32}^{(1)} > \frac{1}{m+1}$, which means the probability of correctly translating a sentence is greater than choosing from potentially wrong translations uniformly at random, cycle dual learning outperforms dual learning.

We also prove a similar theorem for multipath dual learning (see Appendix F). The accuracy of multipath dual learning has a similar form. We observe similar empirical performances as well (see Appendix E).

## 5    EXPERIMENTS OF DUAL LEARNING

Since previous works (He et al., 2016; Xia et al., 2017a;b; Wang et al., 2018; Xia et al., 2018) have demonstrated strength of dual learning, we aims at providing some theoretical insights. We choose

WMT2014 English↔Germen translation[1] and MultiUN (Eisele & Chen, 2010) English↔French translation[2] to verify our theoretical analysis for dual learning. For ease of reference, denote English, French and German as En, Fr and De respectively.

## 5.1 SETTINGS

**Datasets** Following the common practice in NMT, for the En↔De, we preprocess the data in the same way as that used in Vaswani et al. (2017) and eventually obtain $4.5M$ training sentence pairs. We concatenate newstest2012 and newstest2013 as the validation set ($6K$ sentence pairs) and choose newstest2014 as the test set ($3K$ sentence pairs). For the MultiUN En↔Fr translation, we sample $2M/6K/3K$ sentence pairs as the training/validation/test sets. All the sentence are split into word-piece following (Johnson et al., 2016). To leverage dual learning as that used in (He et al., 2016), for WMT'14 En↔De translation, we choose $8M$ monolingual English sentences and $8M$ monolin-gial German sentences from newscrawl. For MultiUN En↔Fr translation, we randomly sample $1M$ English and $1M$ French sentences as the monolingual data to construct the duality loss.

**Architecture & Optimization** We use *Transformer* (Vaswani et al., 2017), the state-of-the-art NMT system, for each direction. For WMT En↔De translation, we choose the *transformer_big* configu-ration, in which the word embedding dimension, hidden dimension filter sizes and number of multi-head attention are 1024, 1024 and 4096 respectively. For MultiUN En↔Fr translation, we choose the *transformer_base* configuration, in which the aforementioned four numbers are 512, 512, 2048 and 8 respectively. The optimization algorithm is Adam with learning rate $2 \times 10^{-4}$, $\beta_1 = 0.99$ and 7168 tokens per GPU. We train our models on 8 M40 GPUs for 7 days.

**Evaluation** For WMT 2014 En↔De translation, following Vaswani et al. (2017), we use beam search with beam width 4 and length penalty 0.6 to generate candidates. The evaluation metric is BLEU score (Papineni et al., 2002b), which is a geometric mean of $n$-gram precisions ($n = 1, 2, 3, 4$). A large BLEU score indicates a better translation quality.

## 5.2 RESULTS

**Translation qualities** The BLEU scores of each translation tasks are summarized in Table 1, in which the second row and third row represent the results of the standard Transformer and dual learning. We can see that after applying dual learning, the performances of all tasks are boosted. Specially, on En→De and De→En translations, we can improve the baseline from 28.40 to 29.97, and from 32.15 to 35.16. On the other task, dual learning can achieve 0.65 and 0.86 point improve-ment, which demonstrates its effectiveness. We found that on MultiUN, we do not achieve as much improvement as WMT. The reason is that MultiUN dataset is a collection of translated documents from the United Nations, which are usually of formal and simple patterns that are easy to learn. As a result, introducing more data might not increase the BLEU so much.

|  | En→De | De→En | En→Fr | Fr→En |
|---|---|---|---|---|
| Standard | 28.40 | 32.04 | 50.26 | 50.56 |
| Dual Learning | 29.97 | 34.93 | 50.91 | 51.42 |

Table 1: BLEU scores of WMT2014 En↔De and MultiUN En↔Fr translations tasks.

**The translator accuracy** We interpret the results in terms of accuracy when different thresholds are selected. We vary the threshold BLEU score from 10 to 40, and the accuracy of each translator is shown in Table 2. Let $S_1$ and $S_2$ denote English and German respectively for the En↔De task (English and French respectively for the En↔Fr task). Values are percentages of translations that are above the threshold.

Qualitatively, we observe that dual learning outcomes are better than single transformers. More inter-esting observations lie in the following quantitative analysis.

---

[1] Data available at `http://www.statmt.org/wmt14/translation-task.html`.
[2] Data available at `http://opus.nlpl.eu/MultiUN.php`

| Threshold BLEU | En↔De | | | | En↔Fr | | | |
|---|---|---|---|---|---|---|---|---|
| | $p_{12}$ | $p_{21}$ | $p_{12}^d$ | $p_{21}^d$ | $p_{12}$ | $p_{21}$ | $p_{12}^d$ | $p_{21}^d$ |
| 10 | 0.42 | 0.55 | 0.66 | 0.75 | 0.82 | 0.80 | 0.82 | 0.81 |
| 20 | 0.27 | 0.38 | 0.55 | 0.67 | 0.77 | 0.74 | 0.78 | 0.75 |
| 30 | 0.13 | 0.21 | 0.38 | 0.50 | 0.67 | 0.64 | 0.68 | 0.66 |
| 40 | 0.06 | 0.10 | 0.24 | 0.32 | 0.55 | 0.53 | 0.56 | 0.55 |

Table 2: Accuracy of translators using different threshold BLEU scores.

**The roles of** $m$ **and** $\gamma$ It appears that all theories throughout this paper are independent of the selection of threshold score. However, in practice, different thresholds result in different $m$ and $\gamma$ values. A higher threshold BLEU score usually means there are more incorrect possible translations, and it will be harder reconstruct the original sentence well. Despite of this, we assume $m$ and $\gamma$ are constants and performed a regression analysis using equation 6, where $m$ and $\gamma$ are independent variables. we compute $p_{12}$, $p_{21}$, and $p_{12}^d$ for all integer threshold scores between 1 and 100, and fit equation 6 by minimizing the absolute value of differences between the predicted $p_{12}^d$ and true $p_{12}^d$, resulting in $m = 92$ and $\gamma = 0.45$ for the En↔De task ($m = 12$ and $\gamma = 0.57$ for the En↔Fr task). The high $\gamma$ value on one hand implies that machine translation is generally a hard task, and on the other hand means there is still space to improve.

## 6 EXPERIMENTS OF MULTIPATH DUAL LEARNING

To verify the effectiveness of cycle dual learning and multipath dual learning, we focus on the translation between English (En), French (Fr) and Spanish (Es). Again, we choose the MultiUN dataset to build the translation models since any two of the aforementioned three languages have bilingual sentence pairs. We study two different settings, where for each language pair, we are provided with $2M$ or $0.2M$ bilingual sentence pairs. For both settings, we choose $1M$ monolingual sentences for each language. We use *transformer_base* for all experiments in this section, where the model is a six-block network, with word embedding size, hidden state size and filter size $512$, $512$ and $2048$. The training process is the same as that in Section 5.

| | En→Fr | Fr→En | En→Es | Es→En | Es→Fr | Fr→Es |
|---|---|---|---|---|---|---|
| Baseline | 50.26 | 50.56 | 55.15 | 55.23 | 47.75 | 48.13 |
| Dual Learning | 50.91 | 51.42 | 55.51 | 55.77 | 48.23 | 48.52 |
| Cycle | 51.28 | 51.89 | 55.97 | 56.17 | 48.62 | 48.87 |

Table 3: Experimental Results on MultiUN (2M bilingual data)

The experimental results of using $2M$ bilingual data and $1M$ monolingual data are shown in Table 3. We can see that on average, dual learning can boost the six baselines (i.e., standard transformer) by $0.55$ point. Although dual learning can achieve verify high scores on MultiUN translation tasks, our proposed cycle dual learning can still improve it by $0.41$ point on average.

| | En→Fr | Fr→En | En→Es | Es→En | Es→Fr | Fr→Es |
|---|---|---|---|---|---|---|
| Baseline | 43.12 | 43.26 | 49.28 | 47.80 | 41.47 | 41.21 |
| Dual Learning | 45.54 | 45.44 | 51.07 | 50.31 | 42.81 | 42.57 |
| Cycle | 47.23 | 46.72 | 52.56 | 51.65 | 43.52 | 44.97 |

Table 4: Experimental Results on MultiUN (0.2M bilingual data)

The results of using 0.2M bilingual data plus 1M monolingual data is shown in Table 4. We have the following observations: (1) Since there are fewer bilingual sentences, the baselines of the six translation tasks are not as good as those in Table 3. (2) For this setting, dual learning can improve the BLEU scores by 1.93 points on average, which is consistent with the discovery in He et al.

(2016) that dual learning can obtain more improvements when the number of bilingual sentences is small.(3) When cycle dual learning is added to the conventional dual learning, we can achieve an extra $1.45$ improvements.

## 7 CONCLUSIONS

We provide the first theoretical study of dual learning and characterize conditions when dual learning outperforms baselines. We also propose two algorithmic extensions of dual learning, the cycle dual learning framework and multipath dual learning framework, which are provably better than dual learning under mild conditions. Our dual learning experiments demonstrate the efficacy of dual learning w.r.t. accuracy and provide insights on the potential power of dual learning. Our cycle dual learning framework achieves a new state-of-the-art BLEU score.

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

## A NOTATIONS

| | |
|---|---|
| $k$ | number of language spaces |
| $S_i$ | $i$-th language space |
| $\mu^{(i)}$ | distribution of sentences in $S_i$ |
| $X^{(i)}$ | the random variable that follows $\mu^{(i)}$ |
| $x^{(i)}$ | one sample (sentence) in $S_i$ |
| $x^{(j)}$ | the correct translation of $x^{(i)}$ in $S_J$ |
| $T_{ij}$ | the baseline translatorthat translates from $S_i$ to $S_j$ |
| $p_{ij}$ | accuracy of the $T_{ij}$ |
| $T_{ij}^d$ | translator that translates from $S_i$ to $S_j$ trained using dual learning |
| $p_{ij}^d$ | accuracy of $T^d{ij}$ |
| $T_{ij}^c$ | translator that translates from $S_i$ to $S_j$ trained using cycle dual learning |
| $p_{ij}^c$ | accuracy of $T_{ij}^c$ |
| $T_{ij}^m$ | translator that translates from $S_i$ to $S_j$ trained using multipath dual learning |
| $p_{ij}^m$ | accuracy of $T_{ij}^m$ |

Table 5: Notations

## B PROOF OF THEOREM 1

*Proof.* Consider a random sample $x^{(1)}$ and the translation from $x^{(1)} \in S_1$ to $S_2$. Before dual learning, the accuracy is $p_{12}$. We analyze the two cases defined earlier in this section.

**Case 1.** $T_{21}(T_{12}(x^{(1)})) = x^{(1)}$. Case 1 consists of two subcases:

Case 1.1: $T_{12}(x^{(1)}) = x^{(2)}$;
Case 1.2: $T_{12}(x^{(1)}) \neq x^{(2)}$.

Although Case 1.2 is not desired, dual learning does not detect it. From equation 2, the probability of Case 1.1 is $\Pr(T_{12}(x^{(1)}) = x^{(2)}, T_{21}(T_{12}(x^{(1)})) = x^{(1)}) = p_{12}p_{21}^r$, According to Assumption 1, we have

$$\Pr(T_{12}(x^{(1)}) \neq x^{(2)}, T_{21}(T_{12}(x^{(1)})) = x^{(1)})$$
$$= \sum_{y^{(2)} \in S_2, y^{(2)} \neq x^{(2)}} \Pr(T_{12}(x^{(1)}) = y^{(2)}, T_{21}(y^{(2)}) = x^{(1)}) = \frac{(1 - p_{12})(1 - p_{21}^r)}{m}$$

**Case 2.** $T_{21}(T_{12}(x^{(1)})) \neq x^{(1)}$. Dual learning will train the translators so that this case is minimized. The probability of this case is simply the complement of Case 1.

$$\Pr(\text{Case 2}) = 1 - p_{12}p_{21}^r - \frac{(1 - p_{12})(1 - p_{21}^r)}{m}$$

After dual learning, Case 2 is redistributed to the two subcases of Case 1, with probabilities $\alpha$ and $\beta$ respectively. So we have

$$\Pr(T_{12}^d(x^{(1)}) = x^{(2)}, T_{21}^d(T_{12}^d(x^{(1)})) = x^{(1)}) = p_{12}p_{21}^r + \alpha(1 - p_{12}p_{21}^r - \frac{(1 - p_{12})(1 - p_{21}^r)}{m})$$

$$= (1 - \alpha)p_{12}p_{21}^r + \alpha(1 - \frac{(1 - p_{12})(1 - p_{21}^r)}{m}) \tag{11}$$

$$\Pr(T_{12}^d(x^{(1)}) \neq x^{(2)}, T_{21}^d(T_{12}^d(x^{(1)})) = x^{(1)}) = \frac{(1 - \beta)(1 - p_{12})(1 - p_{21}^r)}{m} + \beta(1 - p_{12}p_{21}^r)$$

$$\Pr(T_{21}^d(T_{12}^d(x^{(1)})) \neq x^{(1)}) = \gamma(1 - p_{12}p_{21}^r - \frac{(1 - p_{12})(1 - p_{21}^r)}{m}),$$

where equation 11 computes the accuracy of dual learning. □

## C    PROOF OF THEOREM 2

*Proof.* We focus on the mapping from $x^{(1)} \in S_1$ to $S_2$ and consider the following two cases:

Case 1: $T_{31}^d(T_{23}^d(T_{12}^d(x^{(1)}))) = x^{(1)}$;
Case 2: $T_{31}^d(T_{23}^d(T_{12}^d(x^{(1)}))) \neq x^{(1)}$.

We first characterize the probability of Case 1.

**Case 1:** According to whether the translation in $S_2$ is correct, Case 1 can be partitioned into two subcases:

Case 1.1: $T_{12}^d(x^{(1)}) = x^{(2)}, T_{31}^d(T_{23}^d(x^{(2)})) = x^{(1)}$;

Case 1.2: $T_{12}^d(x^{(1)}) \neq x^{(2)}, T_{31}^d(T_{23}^d(T_{12}^d(x^{(1)}))) = x^{(1)}$.

**Case 1.1.** For the path $S_1 \leftrightarrow S_2$, we have $\Pr(T_{12}^d(x^{(1)}) = x^{(2)}) = q_{12}$. For the path $S_2 \leftrightarrow S_3 \leftrightarrow S_1$, since we only care about the translation in $S_1$, we have

$$
\begin{aligned}
\Pr(T_{31}^d(T_{23}^d(x^{(2)})) = x^{(1)}) =\ & \Pr(T_{23}^d(x^{(2)}) = x^{(3)}) \Pr(T_{31}^d(T_{23}^d(x^{(2)})) = x^{(1)} | T_{23}^d(x^{(2)}) = x^{(3)}) \\
& + \Pr(T_{23}^d(x^{(2)}) \neq x^{(3)}) \Pr(T_{31}^d(T_{23}^d(x^{(2)})) = x^{(1)} | T_{23}^d(x^{(2)}) \neq x^{(3)}) \\
=\ & q_{23} q_{31}^{(1)} + \frac{(1 - q_{23})(1 - q_{31}^{(1)})}{m}
\end{aligned}
$$

where the first equality is obtained by the law of total probability, and the second quality is obtained by equation 9 and Assumption 2. Then the probability of Case 1.1 is computed as follows.

$$
\Pr(T_{12}^d(x^{(1)}) = x^{(2)}, T_{23}^d(T_{13}^d(x^{(1)})) = x^{(2)}) = q_{12}(q_{23} q_{31}^{(1)} + \frac{(1 - q_{23})(1 - q_{31}^{(1)})}{m}) \tag{12}
$$

**Case 1.2.** For the $S_1 \leftrightarrow S_2$ path we have $\Pr(T_{12}^d(x^{(1)}) \neq x^{(2)}) = 1 - q_{12}$. Let $y^{(2)} \in S_2$ denote this incorrect translation. Now we compute $\Pr(T_{31}^d(T_{23}^d(x^{(2)})) = y^{(1)})$. There are three cases depending on the translation at $S_3$: $x^{(3)}$, $y^{(3)}$, and others. They are computed as follows.

$$
\Pr(T_{31}^d(T_{23}^d(y^{(2)})) = x^{(1)} | T_{23}^d(x^{(2)}) = x^{(3)}) = q_{31}^{(1)}
$$

$$
\Pr(T_{31}^d(T_{23}^d(y^{(2)})) = x^{(1)} | T_{23}^d(x^{(2)}) = y^{(3)}) = \frac{1 - q_{31}^{(1)}}{m}
$$

$$
\Pr(T_{31}^d(T_{23}^d(y^{(2)})) = x^{(1)} | T_{23}^d(x^{(2)}) \neq x^{(3)}, y^{(3)}) = \frac{1 - q_{31}^{(1)}}{m}
$$

Using the law of total probability we have

$$
\begin{aligned}
\Pr(T_{31}^d(T_{23}^d(y^{(2)})) = x^{(1)}) =\ & \frac{q_{31}^{(1)}(1 - q_{23})}{m} + \frac{q_{23}(1 - q_{31}^{(1)})}{m} + \frac{(m - 1)(1 - q_{23})(1 - q_{31}^{(1)})}{m^2} \\
=\ & \frac{m(q_{32} + q_{31}^{(1)} - 2q_{23}q_{31}^{(1)}) + (m - 1)(1 - q_{23})(1 - q_{31}^{(1)})}{m^2} \\
=\ & \frac{q_{23} + q_{31}^{(1)} - (m + 1)q_{23}q_{31}^{(1)} + m - 1}{m^2}
\end{aligned}
$$

Then the probability of Case 1.2 is

$$
\Pr(T_{12}^d(x^{(1)}) \neq x^{(2)}, T_{31}^d(T_{23}^d(T_{12}^d(x^{(1)}))) = x^{(1)}) = \frac{(1 - q_{12})(q_{23} + q_{31}^{(1)} - (m + 1)q_{32}q_{31}^{(1)} + m - 1)}{m^2} \tag{13}
$$

**Case 2.** The probability of Case 2 is simply the complement of Case 1.

$$
\begin{aligned}
& \Pr(T_{31}^d(T_{23}^d(T_{12}^d(x^{(1)}))) \neq x^{(1)}) = 1 - \Pr(T_{31}^d(T_{23}^d(T_{12}^d(x^{(1)}))) = x^{(1)}) \\
=\ & 1 - q_{12}(q_{23}q_{31}^{(1)} + \frac{(1 - q_{23})(1 - q_{31}^{(1)})}{m}) - \frac{(1 - q_{12})(q_{23} + q_{31}^{(1)} - (m + 1)q_{23}q_{31}^{(1)} + m - 1)}{m^2}.
\end{aligned} \tag{14}
$$

Then the accuracy of this triple learning is

$$p_{12}^m = \Pr(T_{12}^d(x^{(1)}) = x^{(2)}, T_{31}^d(T_{23}^d(x^{(2)})) = x^{(1)}) + \alpha' \Pr(T_{31}^d(T_{23}^d(T_{12}^d(x^{(1)}))) \neq x^{(1)}), \quad (15)$$

where $\alpha'$ is defined in equation 8. equation 10 is obtained by substitute equation 12 and equation 14 into equation 15. $\qquad \square$

## D   MULTIPATH DUAL LEARNING FRAMEWORK

The algorithm of multipath dual learning is shown in Algorithm 2.

---

**Algorithm 2:** Multi-path Dual Learning Framework

1 *Input*: Samples from spaces $S_1 \ldots S_k$, initial translators $T_{12}, T_{21}$ and $T_{1i}, T_{i1} \ \forall i = 3, \ldots, K$; learning rates $\eta$;
2 Train each of $T_{12}, T_{21}$ and $T_{1i}, T_{i1} \ \forall i = 3, \ldots, k$ by dual learning;
3 Randomly sample a $k$ from $\{3, 4, \cdots, K\}$; randomly sample one $x^{(1)} \in S_1$ and one $x^{(2)} \in S_2$;
4 Sample one $\hat{x}^{(k)} \sim T_{1k}(x^{(1)})$, $\hat{x}^{(2)} \sim T_{k2}(\hat{x}^{(k)})$; $\tilde{x}^{(k)} \sim T_{2k}(x^{(2)})$ and $\hat{x}^{(2)} \sim T_{k2}(\tilde{x}^{(k)})$;
5 Update the parameters of $T_{12}$ and $T_{21}$, denoted as $\theta_{12}$ and $\theta_{21}$, as follows:

$$\theta_{12} \leftarrow \theta_{12} - \eta \nabla_{\theta_{12}} \log P(\hat{x}^{(2)} | x^{(1)}; \theta_{12}); \quad \theta_{21} \leftarrow \theta_{21} - \eta \nabla_{\theta_{21}} \log P(\hat{x}^{(1)} | x^{(2)}; \theta_{21}); \quad (16)$$

Repeat Step 3 to Step 5 until convergence;

---

It is well-known that the two gradients in equation 16 are unbiased estimators of

$$\nabla_{\theta_{12}} D_{\mathrm{KL}}(P(\cdot | x^{(1)}; T_{k2} \circ T_{1k}) \| P(\cdot | x^{(1)}; \theta_{12})) \; ; \nabla_{\theta_{21}} D_{\mathrm{KL}}(P(\cdot | x^{(1)}; T_{k2} \circ T_{1k}) \| P(\cdot | x^{(1)}; \theta_{12})). \tag{17}$$

The reason is shown as follows. For any given $x^{(1)} \in S_1$,

$$\min_{\theta_{12}} D_{\mathrm{KL}}(P(\cdot | x^{(1)}; \theta_{1k}, \theta_{k2}) \| P(\cdot | x^{(1)}; \theta_{12}))$$

$$= \sum_{x^{(2)} \in S_2} P(x^{(2)} | x^{(1)}; \theta_{1k}, \theta_{k2}) \log \frac{P(x^{(2)} | x^{(1)}; \theta_{1k}, \theta_{k2})}{P(x^{(2)} | x^{(1)}; \theta_{12})}, \tag{18}$$

which is equivalent to

$$\max_{\theta_{12}} \sum_{x^{(2)} \in S_2} P(x^{(2)} | x^{(1)}; \theta_{1k}, \theta_{k2}) \log P(x^{(2)} | x^{(1)}; \theta_{12})$$

$$= \sum_{x^{(2)} \in S_2} \sum_{x^{(k)} \in S_k} P(x^{(2)}, x^{(k)} | x^{(1)}; \theta_{1k}, \theta_{k2}) \log P(x^{(2)} | x^{(1)}; \theta_{12})$$

$$= \sum_{x^{(2)} \in S_2} \sum_{x^{(k)} \in S_k} P(x^{(k)} | x^{(1)}; \theta_{1k}, \theta_{k2}) P(x^{(2)} | x^{(k)}, x^{(1)}; \theta_{1k}, \theta_{k2}) \log P(x^{(2)} | x^{(1)}; \theta_{12}) \tag{19}$$

$$= \sum_{x^{(2)} \in S_2} \sum_{x^{(k)} \in S_k} P(x^{(k)} | x^{(1)}; \theta_{1k}) P(x^{(2)} | x^{(k)}; \theta_{k2}) \log P(x^{(2)} | x^{(1)}; \theta_{12})$$

$$= \mathbb{E}_{x^{(k)} \sim P(\cdot | x^{(1)}; \theta_{1k})} \mathbb{E}_{x^{(2)} \sim P(\cdot | x^{(k)}; \theta_{k2})} \log P(x^{(2)} | x^{(1)}; \theta_{12}).$$

Therefore, we can first sample $\hat{x}^{(k)}$ from $P(\cdot | x^{(1)}; \theta_{12})$, then sample $\hat{x}^{(2)}$ from $P(\cdot | \hat{x}^{(k)}; \theta_{k2})$, and then maximize $\log P(\hat{x}^{(2)} | x^{(1)}; \theta_{12})$. Obviously, $-\nabla_{\theta_{12}} \log P(\hat{x}^{(2)} | x^{(1)}; \theta_{12})$ is the unbiased estimator of $\nabla_{\theta_{12}} D_{\mathrm{KL}}(P(\cdot | x^{(1)}; \theta_{1k}, \theta_{k2}) \| P(\cdot | x^{(1)}; \theta_{12}))$.

The performance of multipath dual learning is shown in Table 6 in comparison with other translators. We observe that multipath dual learning has a very similar performance with cycle dual learning, which is consistent with our theoretical analysis.

| | En→Fr | Fr→En | En→Es | Es→En | Es→Fr | Fr→Es |
|---|---|---|---|---|---|---|
| Baseline | 50.26 | 50.56 | 55.15 | 55.23 | 47.75 | 48.13 |
| Dual Learning | 50.91 | 51.42 | 55.51 | 55.77 | 48.23 | 48.52 |
| Multi-path | 51.27 | 51.74 | 55.89 | 56.06 | 48.34 | 48.76 |
| Cycle | 51.28 | 51.89 | 55.97 | 56.17 | 48.62 | 48.87 |

Table 6: Experimental Results on MultiUN (2M bilingual data)

## E   DERIVATION OF CYCLE DUAL LEARNING ALGORITHM

We follow the notations used in Appendix D. Let $\mathcal{R}(x^{(2)})$ denote the event that after passing $S_2 \to S_k \to S_1 \to S_2$, $x^{(2)}$ is reconstructed to $x^{(2)}$. We have that

$$
\begin{aligned}
&\log P(\mathcal{R}(x^{(2)})) = \log P(x^{(2)}|\text{start from } x^{(2)}; \theta_{2k}, \theta_{k1}, \theta_{12}) \\
&= \sum_{x^{(k)} \in S_k} \sum_{x^{(1)} \in S_1} \log P(x^{(2)}, x^{(1)}, x^{(k)}|\text{start from } x^{(2)}; \theta_{2k}, \theta_{k1}, \theta_{12}) \\
&= \sum_{x^{(k)} \in S_k} \sum_{x^{(1)} \in S_1} \log P(x^{(1)}, x^{(k)}|\text{start from } x^{(2)}; \theta_{2k}, \theta_{k1}) \cdot \\
&\qquad\qquad P(x^{(2)}|\text{start from } x^{(2)}, x^{(1)}, x^{(k)}; \theta_{2k}, \theta_{k1}, \theta_{12}) \\
&\geq \sum_{x^{(k)} \in S_k} \sum_{x^{(1)} \in S_1} P(x^{(1)}, x^{(k)}|x^{(2)}; \theta_{2k}, \theta_{k1}) \log P(x^{(2)}|\text{start from } x^{(2)}, x^{(1)}, x^{(k)}; \theta_{2k}, \theta_{k1}, \theta_{12}) \\
&= \sum_{x^{(k)} \in S_k} \sum_{x^{(1)} \in S_1} P(x^{(1)}, x^{(k)}|x^{(2)}; \theta_{2k}, \theta_{k1}) \log P(x^{(2)}|x^{(1)}; \theta_{12}) \\
&= \sum_{x^{(k)} \in S_k} \sum_{x^{(1)} \in S_1} P(x^{(k)}|x^{(2)}; \theta_{2k}) P(x^{(1)}|x^{(k)}; \theta_{k1}) \log P(x^{(2)}|x^{(1)}; \theta_{12}) \\
&= \mathbb{E}_{x^{(k)} \sim P(\cdot|x^{(2)}; \theta_{2k})} \mathbb{E}_{x^{(1)} \sim P(\cdot|x^{(k)}); \theta_{k1}} \log P(x^{(2)}|x^{(1)}; \theta_{12}).
\end{aligned}
$$
(20)

Then, we can use the algorithm proposed in Algorithm 1 to solve the above optimization problem, i.e., $\min -\log P(\mathcal{R}(x^{(2)}))$.

## F   THEORETICAL ANALYSIS FOR MUTIPATH DUAL LEARNING

We define $q_{32}^{(1)} = \Pr_{X^{(3)} \sim T_{13}^d(X^{(1)})}(T_{32}^d(x^{(3)}) = x^{(2)})$, which as the accuracy of $T_{32}^d$ when the sentence in $S_3$ is translated from a random sentence in $S_1$. This is analogous to reconstruction accuracy, and is reduced to the reconstruction accuracy when $S_2 = S_1$. To analyze multipath learning, we consider the following cases (analogous to Case 1 and Case 2 for dual learning and cycle dual learning): **Case 1:** $T_{12}(x^{(1)}) = T_{32}(T_{13}(x^{(1)}))$; **Case 2:** $T_{12}(x^{(1)}) \neq T_{32}(T_{13}(x^{(1)}))$.

multipath dual learning will detect Case 2 and train the translators so that the difference between translations from the two paths is minimized. To quantify this effect, we define the following probabilities:

$$
\begin{aligned}
\alpha' &= \Pr(T_{12}^m(x^{(1)}) = x^{(2)}, T_{32}^m(T_{13}^m(x^{(1)})) = x^{(2)}|\text{Case 2}) \\
\beta' &= \Pr(T_{12}^m(x^{(1)}) \neq x^{(2)}, T_{32}^m(T_{13}^m(x^{(1)})) = T_{12}^m(x^{(2)})|\text{Case 2}) \\
\gamma' &= \Pr(T_{21}^m(T_{12}^m(x^{(1)})) \neq x^{(1)}|\text{Case 2}),
\end{aligned}
$$
(21)

where Case 2 denotes the condition $T_{12}(x^{(1)}) \neq T_{32}(T_{13}(x^{(1)}))$ and superscripts $d$ and $m$ denote the translators obtained from dual learning and multipath learning, respectively. $\alpha', \beta', \gamma'$ can be viewed as the probability of correcting the wrong translations by multipath learning, the probability of the occurrence of the alignment problem under Case 2, and the probability of nonzero reconstruction error. $\gamma'$ models the imperfectness of dual learning. And again, we have $\alpha' + \beta' + \gamma' = 1$. We

assume that $T_{13}^d$ and $T_{32}^d$ are independent translators before multipath learning. That implies,

$$\Pr(T_{13}^d(x^{(1)}) = x^{(3)}, T_{32}^d(T_{13}^d(x^{(1)})) = x^{(2)}) = \Pr(T_{13}^d(x^{(1)}) = x^{(3)}, T_{32}^d(x^{(3)}) = x^{(2)}) = q_{13}q_{32}^{(1)}$$
(22)

Then we have the following theorem, which focuses one the triangle structure containing only $S_1$, $S_2$, and $S_3$. The more general case can be viewed as adding one path a time so Theorem 3 can be applied.

**Theorem 3.** *Given languages spaces $S_1$, $S_2$, and $S_3$, where the objective is to train a translator that maps from $S_1$ to $S_2$, the accuracy of two-path learning (Figure 1) outcome is*

$$p_{12}^m = q_{12}(1 - \alpha')(q_{13}q_{32}^{(1)} + \frac{(1 - q_{13})(1 - q_{32}^{(1)})}{m})$$
$$+ \alpha'(1 - \frac{(1 - q_{12})(q_{13} + q_{32}^{(1)} - (m + 1)q_{13}q_{32}^{(1)} + m - 1)}{m^2})$$
(23)

*under Assumption 2.*

*Proof.* Similar to the analysis of Theorem 1, we focus on the mapping from $x^{(1)} \in S_1$ to $S_2$ and consider the following two cases:

Case 1: $T_{12}^d(x^{(1)}) = T_{32}^d(T_{13}^d(x^{(1)}))$;
Case 2: $T_{12}^d(x^{(1)}) \neq T_{32}^d(T_{13}^d(x^{(1)}))$.

We first characterize the probability of Case 1.

**Case 1:** According to whether the translation in $S_2$ is correct, Case 1 can be partitioned into two subcases:

Case 1.1: $T_{12}^d(x^{(1)}) = x^{(2)}, T_{23}^d(T_{13}^d(x^{(1)})) = x^{(2)}$;

Case 1.2: $T_{12}^d(x^{(1)}) \neq x^{(2)}, T_{23}^d(T_{13}^d(x^{(1)})) = T_{12}^d(x^{(1)})$.

**Case 1.1.** For the path $S_1 \leftrightarrow S_2$, we have

$$\Pr(T_{12}^d(x^{(1)}) = x^{(2)}) = q_{12}.$$

For the path $S_1 \leftrightarrow S_3 \leftrightarrow S_2$, since we only care about the translation in $S_2$, we have

$$\Pr(T_{32}^d(T_{13}^d(x^{(1)})) = x^{(2)})$$
$$= \Pr(T_{13}^d(x^{(1)}) = x^{(3)}) \Pr(T_{32}^d(T_{13}^d(x^{(1)})) = x^{(2)} | T_{13}^d(x^{(1)}) = x^{(3)})$$
$$+ \Pr(T_{13}^d(x^{(1)}) \neq x^{(3)}) \Pr(T_{32}^d(T_{13}^d(x^{(1)})) = x^{(2)} | T_{13}^d(x^{(1)}) \neq x^{(3)})$$
$$= q_{13}q_{32}^{(1)} + \frac{(1 - q_{13})(1 - q_{32}^{(1)})}{m}$$

where the first equality is obtained by the law of total probability, and the second quality is obtained by equation 22 and Assumption 2. Then the probability of Case 1.1 is computed as follows.

$$\Pr(T_{12}^d(x^{(1)}) = x^{(2)}, T_{23}^d(T_{13}^d(x^{(1)})) = x^{(2)}) = q_{12}(q_{13}q_{32}^{(1)} + \frac{(1 - q_{13})(1 - q_{32}^{(1)})}{m})$$
(24)

**Case 1.2.** For the $S_1 \leftrightarrow S_2$ path we have $\Pr(T_{12}^d(x^{(1)}) \neq x^{(2)}) = 1 - q_{12}$. Let $y^{(2)} \in S_2$ denote this incorrect translation. Now we compute $\Pr(T_{32}^d(T_{13}^d(x^{(1)})) = y^{(2)})$. There are three cases depending on the translation at $S_3$: $x^{(3)}, y^{(3)}$, and others. They are computed as follows.

$$\Pr(T_{32}^d(T_{13}^d(x^{(1)})) = y^{(2)} | T_{13}^d(x^{(1)}) = x^{(3)}) = \frac{1 - q_{32}^{(1)}}{m}$$
$$\Pr(T_{32}^d(T_{13}^d(x^{(1)})) = y^{(2)} | T_{13}^d(x^{(1)}) = y^{(3)}) = q_{32}^{(1)}$$
$$\Pr(T_{32}^d(T_{13}^d(x^{(1)})) = y^{(2)} | T_{13}^d(x^{(1)}) \neq x^{(3)}, y^{(3)}) = \frac{1 - q_{32}^{(1)}}{m}$$

Using the law of total probability we have

$$
\begin{aligned}
\Pr(T_{32}^d(T_{13}^d(x^{(1)})) = y^{(2)}) =& \frac{q_{13}(1 - q_{32}^{(1)})}{m} + \frac{q_{32}^{(1)}(1 - q_{13})}{m} + \frac{(m-1)(1 - q_{13})(1 - q_{32}^{(1)})}{m^2} \\
=& \frac{m(q_{13} + q_{32}^{(1)} - 2q_{13}q_{32}^{(1)}) + (m-1)(1 - q_{13})(1 - q_{32}^{(1)})}{m^2} \\
=& \frac{q_{13} + q_{32}^{(1)} - (m+1)q_{13}q_{32}^{(1)} + m - 1}{m^2}
\end{aligned}
$$

Then the probability of Case 1.2 is

$$
\begin{aligned}
& \Pr(T_{12}^d(x^{(1)}) \neq x^{(2)}, T_{23}^d(T_{13}^d(x^{(1)})) = T_{12}^d(x^{(1)})) \\
=& \frac{(1 - q_{12})(q_{13} + q_{32}^{(1)} - (m+1)q_{13}q_{32}^{(1)} + m - 1)}{m^2}
\end{aligned}
\tag{25}
$$

**Case 2.** The probability of Case 2 is simply the complement of Case 1.

$$
\begin{aligned}
& \Pr(T_{12}^d(x^{(1)}) \neq T_{32}^d(T_{13}^d(x^{(1)}))) = 1 - \Pr(T_{12}^d(x^{(1)}) = T_{32}^d(T_{13}^d(x^{(1)}))) \\
=& 1 - q_{12}(q_{13}q_{32}^{(1)} + \frac{(1 - q_{13})(1 - q_{32}^{(1)})}{m}) - \frac{(1 - q_{12})(q_{13} + q_{32}^{(1)} - (m+1)q_{13}q_{32}^{(1)} + m - 1)}{m^2}.
\end{aligned}
\tag{26}
$$

Then the accuracy of this two-path learning is

$$
p_{12}^m = \Pr(T_{12}^d(x^{(1)}) = x^{(2)}, T_{23}^d(T_{13}^d(x^{(1)})) = x^{(2)}) + \alpha' \Pr(T_{12}^d(x^{(1)}) \neq T_{32}^d(T_{13}^d(x^{(1)}))), \tag{27}
$$

where $\alpha'$ is defined in equation 21. equation 23 is obtained by substitute equation 24 and equation 26 into equation 27. □

**The role of $T_{12}^d$.** The accuracy increases as $q_{12}$ increases. So a good original translator trained from dual learning is desirable. However, if $T_{12}^d$ is arbitrarily bad, it is still possible to improve it significantly. This implies the application of this framework on low-resource or even zero-shot machine translation.

**A similar hypothesis.** Similar to the analysis of dual learning, we consider the condition where $\frac{\alpha'}{\beta'} = \frac{\Pr(T_{12}^d(x^{(1)})=x^{(2)}, T_{23}^d(T_{13}^d(x^{(1)}))=x^{(2)})}{\Pr(T_{12}^d(x^{(1)})\neq x^{(2)}, T_{23}^d(T_{13}^d(x^{(1)}))=x^{(2)})}$ and define $\Gamma' = \gamma' \Pr(T_{12}^d(x^{(1)}) \neq T_{32}^d(T_{13}^d(x^{(1)})))$. Then the accuracy simplifies to

$$
\begin{aligned}
p_{12}^m =& \frac{\alpha'(1 - \Gamma')}{\alpha' + \beta'} = \frac{q_{12}(q_{13}q_{32}^{(1)} + \frac{(1-q_{13})(1-q_{32}^{(1)})}{m})(1 - \Gamma')}{q_{12}(q_{13}q_{32}^{(1)} + \frac{(1-q_{13})(1-q_{32}^{(1)})}{m}) + \frac{(1-q_{12})(q_{13}+q_{32}^{(1)}-(m+1)q_{13}q_{32}^{(1)}+m-1)}{m^2}} \\
=& \frac{1 - \Gamma'}{1 + \frac{(1-q_{12})(q_{13}+q_{32}^{(1)}-(m+1)q_{13}q_{32}^{(1)}+m-1)}{mq_{12}(mq_{13}q_{32}^{(1)}+(1-q_{13})(1-q_{32}^{(1)}))}} = \frac{1 - \Gamma'}{1 + M\frac{1-q_{12}}{q_{12}}}
\end{aligned}
$$

where $M = \frac{q_{13}+q_{32}^{(1)}-(m+1)q_{13}q_{32}^{(1)}+m-1}{m(mq_{13}q_{32}^{(1)}+(1-q_{13})(1-q_{32}^{(1)}))}$. When $\gamma' = 0$ (and therefore $\Gamma' = 0$) and $M = 1$, it simplifies to $q_{12}$, which is the accuracy of dual learning. Observing $p_{12}^m$ increases as $M$ decreases, we let $M < 1$ so that $p_{12}^m > q_{12}$. We have

$$
\frac{q_{13} + q_{32}^{(1)} - (m+1)q_{13}q_{32}^{(1)} + m - 1}{m(mq_{13}q_{32}^{(1)} + (1 - q_{13})(1 - q_{32}^{(1)}))} < 1
$$

$$
((m+1)q_{13} - 1)((m+1)q_{32}^{(1)} - 1) > 0
$$

when $q_{13}, q_{32}^{(1)} > \frac{1}{m+1}$, which means the probability of correctly translating a sentence is greater than choosing from potentially wrong translations uniformly at random, the two-path learning outperforms dual learning.

