# OpenReview forum: "Dual Learning: Theoretical Study and Algorithmic Extensions"
_ICLR.cc/2019/Conference_

### Official Review · AnonReviewer3 · 2018-11-03
**Overlooked assumption?**

**Rating:** 5
**Confidence:** 3

**Review:**

This paper provides a theoretical perspective of the dual learning tasks and proposes two generalizations (multipath/cycle dual learning) that utilize multiple language sets. Through experiments, the paper discusses the relationship between theoretical perspective and actual translation quality.

Overall, the paper is well written and discussed enough. My concern is about Theorem 1 that could be a critical problem.
In the proof of Theorem 1, it discussed that the dual learning can minimize Case 2. This assumption is reasonable if the vanilla translator is completely fixed (i.e., no longer updated) but this condition may not be assumed by the authors as far as I looked at Algorithm 2 and 3 that update the parameters of vanilla translators directly. The proof is constructed by only the effect against Case 2. However, if the vanilla translator is directly updated through dual training, there should exist some random changes in also Case 1 and this behavior should also be included in the theorem.

Correction and suggestions writing:
* It is better to introduce an additional $\alpha$, $\beta$ and $\gamma$ for the vanilla translation accuracy (e.g., $\alpha_0 := p_{ij}p_{ji}^r$) so that most formulations in Section 3 can be largely simplified.
* In Justification of Assumption1 ... "the probability of each cluster is close to $p_max$" ->  maybe "greater than $p_max$" to satisfy the next inequality.
* Eq. (3) ... $T_{ji}^d(T_{ij}^d(x^{(i)})$ -> $T_{ji}^d(x^{(j)})$ to adjust other equations.
* Section 3 Sentence 1: "translatorby" -> "translator by"
* Section 4.2: ${\rm Pr}_{ X^{(3)} \sim T_{23}^d (X^{(1)}) }$ -> $... (X^{(2)}) }$

---

### Official Review · AnonReviewer2 · 2018-11-05
**Possibly some interesting ideas here, but has errors, and also has a very strong assumption (that each sentence has a unique translation)**

**Rating:** 2
**Confidence:** 4

**Review:**

The paper gives theorems concerning "dual learning" - that is, making
use of round-trip consistency in learning of translation and other
tasks.

There are some interesting ideas here. Unfortunately, I think there
are issues with clarity/choice of notation and correctness (errors
resulting from problems with the notation - or at least it's very
hard to figure out if things are correct under some intepretation).

More specifically, I'm uneasy about the use of x^i and x^j as defined
in section 2. In some cases x^j is a deterministic function of x^i, in
some cases it's a random variable, these cases are mixed. Section 2
becomes tangled up in this issue. It would be much better I think to
define a function f_ij(x) for each (i,j) pair that maps a sentence
x \in S^i to its correct translation f_ij(x) \in S^j.

A critical problem with the paper is that Eq. 2 is I think incorrect.
Clearly,

Pr(T_ij(x_i) = f_ij(x_i), T_ji(f_ij(x_i)) = x_i)      [1]
=
Pr(T_ij(x_i) = f_ij(x_i))                             [2]
*
Pr(T_ji(f_ij(x_i)) = x_i) | T_ij(x_i) = f_ij(x_i))    [3]

I think [1] is what is meant by the left-hand-side of Eq 2 in the paper -
though the use of x^j is ambiguous (this ambiguity is a real issue).

It can be verified that

Pr(T_ij(x_i) = f_ij(x_i)) = p_ij

however

Pr(T_ji(f_ij(x_i)) = x_i) | T_ij(x_i) = f_ij(x_i)) \neq p^r_ji

The definition of p^r_ji is that of a different quantity.

This problem unfortunately permeates the statement of theorem 1, and
the proof of theorem 1. It is probably fixable but without a
significantly revised version of the paper a reader/reviewer is
basically guessing what a corrected version of the paper would
be. Unfortunately I think publishing the paper with errors such as
this would be a problem.

Some other points:

[1] The theorem really does have to assume that there is a unique
correct translation f_ij(x^i) for each sentence x^i. Having multiple
possible translations breaks things. The authors say in section 2 "In
practice, we may select a threshold BLEU (Papineni et al., 2002a)
score, above which the translation is considered correct": this seems
to imply that the results apply when multiple translations (above
a certain BLEU score) are possible. But my understanding is that this
will completely break the results (or at least require a significant
modification of the theory).

[2] A further problem with ambiguity/notation is that T^d is never
explicitly defined. Presumably we always have T_ij^s(x^i) = T_ij(x^i) if
T_ji(T_ij(x^i)) = x^i? That needs to be explicitly stated.

[3] There may be something interesting in theorem 1 - putting aside point
[1] above - but I am just really uneasy with this theorem and its proof
given that it uses p^r_ji, and the issue with Eq. 2.

[4] Another issue with the definition of p^r_ij: the notation
P_{X^(j, r) ~ \mu}(...) where the expression ... does not refer
to X^{j, r} (instead it refers to x^j) is just really odd,
and confusing.

---

### Official Review · AnonReviewer1 · 2018-11-06
**Theoretical study and empirical testing of improvement of machine translation classifiers using dual and cyclical paths among languages**

**Rating:** 6
**Confidence:** 3

**Review:**

The paper addresses a means of boosting the accuracy of automatic translators (sentences) by training dual models (a.k.a. language A to B, B to A), multipath (e.g. A to B to C) and cyclical (e.g. A to B to C to A) while starting with well initialized models for translating simple pairs. The idea that additional errors are revealed and allow classifiers to adapt, thus boosting classifier performance, is appealing and intuitive. That a theoretical framework is presented to support this assumption is encouraging, but the assumptions behind this analysis (e.g. Assumption 1) are rather strong. Equation 2 assumes independence. Equations 3-5 can be presented more clearly. Not only are translation errors not uniform in multiclass settings, but they can be highly correlated - this being a possible pitfall of boosting approaches, seen as general classifier errors. The same strong assumptions permeate from the dual to the cyclical case. On the other hand, the (limited) empirical test presented, using a single dataset and baseline classifier method, does support the proposed improvement by boosting, albeit leading to improvements which are slight.

---

### Author Response · Authors · 2018-11-26
**Thank you!**

We thank all reviewers for the very helpful comments and suggestions! We will revise our paper accordingly.

---

### Meta-Review · Area_Chair1 · 2018-12-02
**Reject**

**Confidence:** 4
**Recommendation:** Reject

**Metareview:**

The reviewers vary in their scores but overall there is agreement that this paper is not ready for acceptance.